# Combined administration of anisodamine and neostigmine alleviated colitis by inducing autophagy and inhibiting inflammation

**Mengzhen Liu[1,2]⊙, Danni Zhu[1]⊙, Hui Yan[1]⊙, Zhiwei Dong[3], Jingjing Zhang[1], Ni Kong[1], Guangyu Zhang[1], Qin Xu[1], Ting Han[1], Ping Ke[1,2]\*, Chong Liu[1]\***

**1** Department of Pharmacy, Second Military Medical University/Navy Military Medical University, Shanghai, China, **2** Air Force Hangzhou Special Service Recuperation Center Sanatorium Area 4, Nanjing, China, **3** Department of General Surgery, Air Force Medical Center, PLA, Beijing, China

⊙ These authors contributed equally to this work.
\* wanlc2004@aliyun.com (CL); keping1989@126.com (PK)

**Data Availability Statement:** All relevant data are within the paper and its Supporting information files.

## Abstract

Our previous work demonstrated that the anisodamine (ANI) and neostigmine (NEO) combination produced an antiseptic shock effect and rescued acute lethal crush syndrome by activating the α7 nicotinic acetylcholine receptor (α7nAChR). This study documents the therapeutic effect and underlying mechanisms of the ANI/NEO combination in dextran sulfate sodium (DSS)-induced colitis. Treating mice with ANI and NEO at a ratio of 500:1 alleviated the DSS-induced colitis symptoms, reduced body weight loss, improved the disease activity index, enhanced colon length, and alleviated colon inflammation. The combination treatment also enhanced autophagy in the colon of mice with DSS-induced colitis and lipopolysaccharide/DSS-stimulated Caco-2 cells. Besides, the ANI/NEO treatment significantly reduced INF-γ, TNF-α, IL-6, and IL-22 expression in colon tissues and decreased TNF-α, IL-1β, and IL-6 mRNA levels in Caco-2 cells. Meanwhile, the autophagy inhibitor 3-methyladenine and ATG5 siRNA attenuated these effects. Furthermore, 3-methyladenine (3-MA) and the α7nAChR antagonist methyllycaconitine (MLA) weakened the ANI/NEO-induced protection on DSS-induced colitis in mice. Overall, these results indicate that the ANI/NEO combination exerts therapeutic effects through autophagy and α7nAChR in a DSS-induced colitis mouse model.

## Introduction

Although the exact etiology of inflammatory bowel disease (IBD)—consisting of ulcerative colitis (UC) and Crohn's disease—remains unclear, known pathogenesis and progression factors include genetic susceptibility, commensal microbiota, epithelial barrier function, and immune response [1, 2]. Disturbed microbial homeostasis and intestinal epithelial barrier function trigger the innate and adaptive immune responses to stabilize the intestinal environment. However, an over-activation of the self-protective immune responses can increase proinflammatory cytokines secretion and damage the intestine and intestinal mucosal barrier,

**Funding:** This work was supported by grants from the National Natural Science Foundation of China (81903630 to PK) and Shanghai Biomedical Science and Technology Support Program (20S190280 to CL) in study design, data collection and analysis. It was also supported by National Natural Science Foundation of China (81871596 CL and 81800473 to ZWD) in decision to publish and preparation of the manuscript. There was no additional external funding received for this study.

**Competing interests:** The authors have declared that no competing interests exist.

contributing to the pathogenesis and progression of IBD [3, 4]. Consequently, preventing the over-activation of the innate and adaptive immune responses is a potential IBD treatment strategy.

Autophagy is a self-protecting cellular catabolic pathway involving the degradation and recycling of long-lived proteins, damaged organelles, and misfolded proteins [5, 6]. Autophagy plays a vital role in intestinal homeostasis and regulates the interactions between the host's defense and commensal microbiota or intestinal pathogens [7, 8]. Several inflammatory diseases, such as UC and Crohn's disease, involve an autophagy dysfunction [6]. Our previous works demonstrated that enhancing autophagy hampered the pathogenesis of experimental autoimmune encephalomyelitis (EAE) and dextran sulfate sodium (DSS)-induced colitis by inhibiting inflammation [9, 10]. In addition, we found that combining anisodamine (ANI), a muscarinic cholinergic receptor antagonist, with neostigmine (NEO), a cholinesterase inhibitor, significantly rescued acute lethal crush syndrome and collagen-induced arthritis by reducing inflammation [11, 12]. Based on these studies, we hypothesized that the ANI and NEO combination exerted therapeutic effects by promoting autophagy, balancing the immune response in DSS-induced colitis mice models.

The $\alpha 7$ nicotinic acetylcholine receptor ($\alpha 7$nAChR) is a subtype of the N-acetylcholine receptor, a ligand-gated ion channel [13]. $\alpha 7$nAChR plays an immunomodulatory role in various diseases. We also previously confirmed the protective effect of $\alpha 7$nAChR in many disease models such as ischemic stroke and atherosclerosis models [14, 15]. Moreover, we found that the ANI and NEO combination rescued acute lethal crush syndrome by activating $\alpha 7$nAChR [11]. Herein, we demonstrate that the ANI and NEO combination protects mice from DSS-induced colitis by activating $\alpha 7$nAChR.

## Materials and methods

### Animals and reagents

We purchased C57BL/6 mice from Shanghai Super-B&K Laboratory Animal Corp., Ltd. (Shanghai, China) and housed them at 22°C under a 12-h light/dark cycle with unlimited access to water and a standard rodent diet. We followed the guidelines of the Animal Care Committee of the Second Military Medical University, which approved all the experiments. We obtained ANI from Shanghai First Biochemical Pharmaceutical Company (Shanghai, China) and NEO from San-Wei Pharmaceutical Company (Shanghai, China). We purchased 3-methyladenine (3-MA, Cat.No. M9281) and methyllycaconitine (MLA, Cat.No. M168) from Sigma-Aldrich (Saint Louis, MO, USA).

### Cell culture

We obtained Caco-2 cells from the Cell Bank of the Chinese Academy of Sciences (Shanghai, China) and cultured them with Dulbecco's modified Eagle medium (DMEM) (Gibco, Grand Island, NY, USA) supplemented with 10% (vol/vol) fetal bovine serum (FBS) (Gibco, Grand Island, NY, USA) at 37°C in a humidified incubator with 5% $CO_2$. We primed the Caco-2 cells with 10 ng/ml lipopolysaccharide (LPS) (Sigma, Louis, MO, USA) for 1 h and then stimulated them with 3% DSS and 0, 10, 30, or 100 μM of ANI/NEO for 24 h.

### Induction of colitis

We induced colitis in the C57BL/6 mice by providing them *ad libitum* drinking water containing 3% (w/v) DSS (36,000–50,000 kDa, MP Biomedicals, Solon, OH, USA) for 7 days, as

previously described [9]. We replenished the DSS solution every other day. Control mice received only normal drinking water.

## Clinical score and histological analysis

Two investigators blinded to the treatment groups determined the body weight and disease activity index (DAI) of the mice. The DAI combines scores for diarrhea and the presence of occult or overt blood in the stool. We used a previously published scoring system to assess diarrhea and the presence of occult or overt blood in the stool [16]. We recorded body weight changes as percentages relative to the baseline body. Postmortem, we removed the colon and used pieces of colonic tissue for *ex vivo* analysis. For histology, we fixed rings of the transverse part of the colon in 4% buffered formalin and embedded them in paraffin. We stained sections with hematoxylin and eosin according to standard protocols. A pathologist blinded to the treatment groups performed histological scoring as previously described [16]. Briefly, no evidence of inflammation was scored as 0, low level of inflammation with scattered infiltrating mononuclear cells as 1, moderate inflammation with multiple foci as 2, high level of inflammation with increased vascular density and marked wall thickening as 3, and maximal severity of inflammation with transmural leukocyte infiltration and loss of goblet cells as 4.

## Reverse transcription and real-time polymerase chain reaction (PCR)

We extracted total RNA from colon tissues and Caco-2 cells using TRIzol reagent (Invitrogen, Carlsbad, CA, USA). Next, we reverse-transcribed the extracted RNA into cDNA with Prime-Script™ RT Master Mix (Takara, Otsu, Shiga, Japan). We then performed real-time PCR using a LightCycler quantitative PCR apparatus (Stratagene, Santa Clara, CA, USA) and the FastStart Universal SYBR Green Master (Roche, Konzern-Hauptsitz, Grenzacherstrasse, Switzerland). To eliminate the effect of DSS on PCR, we added 1.25 mM spermine into the PCR reactions. We normalized the expression levels of INF-γ, TNF-α, IL-6, and IL-22 to GAPDH in the same sample and then normalized to the control. After transfection with control small interfering RNA (siRNA) or ATG5 siRNA, we primed Caco-2 cells with 10 ng/ml LPS for 1 h, then stimulated them with 3% DSS in the presence or absence of 100 μM ANI/NEO for 24 h. We normalized the expression levels of TNF-α, IL-1β, and IL-6 to β-actin in the same sample and then normalized to the control. Table 1 lists the primer sequences.

**Table 1. The sequences of the primers used in real-time PCR (RT-PCR) were listed as below.**

|  | Sense (5'-3') | Anti-Sense (5'-3') |
|---|---|---|
| mouse TNF-α | CCCTCCTTCAGACACCCT | GGTTGCCAGCACTTCACT |
| mouse INF-γ | CAGGCCATCAGCAACAACATAAGC | AGCTGGTGGACCACTCGGATG |
| mouse IL-6 | CAATAACCACCCCTGACC | GCGCAGAATGAGATGAGTT |
| mouse IL-22 | GTGAGAAGCTAACGTCCATC | GTCTACCTCTGGTCTCATGG |
| mouse GAPDH | CCCATCACCATCTTCCAGGAG | TTCACCACCTTCTTCTTGATGTCAT |
| human ATG-5 | GCTTCGAGATGTGTGGTTT | GTTCTGCTTCCCTTTCAGTT |
| human TNF-α | CCCTCCTTCAGACACCCT | GGTTGCCAGCACTTCACT |
| human IL-1β | TTGAGTCTGCCCAGTTCC | TTTCTGCTTGAGAGGTGCT |
| human IL-6 | CAATAACCACCCCTGACC | GCGCAGAATGAGATGAGTT |
| human β-actin | CATGTACGTTGCTATCCAGGC | CTCCTTAATGTCACGCACGAT |

## Enzyme-linked immunosorbent assay (ELISA)

We quantified INF-γ and TNF-α in colonic tissue using commercial enzyme-linked immuno-sorbent assay kits (Westang Biotechnology, Shanghai, China).

## Western blotting

We extracted colonic tissue and Caco-2 cells proteins using a standard extraction reagent supplemented with protease inhibitors (Kangchen, Shanghai, China). We quantified the total proteins using a BCA protein assay kit (Beyotime Biotechnology, Shanghai, China). Next, we separated the proteins using SDS-PAGE, electro-transferred them to nitrocellulose membranes, and then incubated them with rabbit anti-Beclin-1 monoclonal antibody (1:500; Cell Signaling Technology, Danvers, MA, USA), rabbit anti-LC3 polyclonal antibody (1:500; Novus Biologicals, Littleton, CO, USA), rabbit anti-p62 antibody (1:500; Cell Signaling Technology, Danvers, MA, USA), mouse anti-beta-actin antibody (1:1000, Beyotime Biotechnology, Shanghai, China), and mouse anti-glyceraldehyde-3-phosphate dehydrogenase (GAPDH) antibody (1:1000, Beyotime Biotechnology, Shanghai, China). Finally, we incubated the membranes with a Donkey anti-rabbit or Donkey anti-mouse secondary antibody (1:5000, LI-COR Biosciences, Lincoln, NE, USA) accordingly. We acquired images with an Odyssey infrared imaging system (Li-Cor Bioscience, Lincoln, NE, USA).

## Autophagy flux assessment

We seeded Caco-2 cells on cultural slides and transfected them with tandem fluorescent AdPlus-mCherry-GFP-LC3B plasmid (Beyotime Biotechnology, Shanghai, China) when the confluence reached 50%–70% [10]. In brief, we cultured the cells in DMEM supplemented with 10% FBS for 24 h, incubated them with plasmids for 8 h, then replaced the medium with DMEM supplemented with 10% FBS for another 48 h to ensure the expression of the genes. After transfection, we primed the cells with 10 ng/ml LPS for 1 h, then stimulated them with 3% DSS and 0 or 100 μM ANI/NEO for 24 h. We detected autophagosomes ($G^+C^+$) and autolysosomes ($G^-C^+$) by confocal microscopy (Leica TCS SP8, Leica, Biberach, Germany) and counted the puncta ($>1$ μm) per cell.

## Transient transfection and siRNA

The following siRNAs against ATG5 (Gene ID: 11793) were synthesized by Genepharm Biotech (Shanghai, China): siRNA1, 5′–GCCUGUAUGUACUGCUUUAUU–3′, 5′–UAAAGCAGUACAUACAGGCUU–3′; siRNA2, 5′–GAACCAUACUAUGCAUUAUU–3′, 5′–AUAAUGCAUAGUAUGGUUCUU–3′; siRNA3, 5′–GGGAAGAAGAGAUUGUUUAUU–3′, 5′–UAAACAAUCUCUUCUUCCCUU–3′. All siRNAs consisted of 21 nucleotides and contained symmetric 3′ overhangs of two deoxythymidines. We transfected the Caco-2 cells with siRNAs as previously reported [17].

## Animals and treatment

To assess the effects of ANI/NEO on DSS-induced colitis, we treated C57BL/6 mice with 3% DSS for seven days and with the vehicle, ANI (20 mg/kg, i.p.), NEO (40 μg/kg, i.p.) or ANI and NEO (ANI/NEO, 5 mg/kg and 10 μg/kg, 10 mg/kg and 20 μg/kg, or 20 mg/kg and 40 μg/kg, i.p.) twice a day from the third to the seventh day. On day 7, mice were euthanized by cervical dislocation after anesthetized with isoflurane. Any mouse appearing a severe weight loss, intractable diarrhea, ruffled fur, or hunched posturea were euthanized immediately. We determined body weight, DAI, colon length, and histologic score as described above. In addition,

we determined the LC3-II/LC3-I ratio; Beclin-1 and P62 protein levels; and INF-γ, TNF-α, IL-6, and IL-22 mRNA levels in colonic tissue as described above.

To assess the influence of autophagy and 7nAChR on the protective effect of ANI/NEO in DSS-induced colitis. We treated C57BL/6 mice with 3% DSS for 7 days and with the vehicle, 3-MA (10 mg/kg, i.p.), MLA (10 mg/kg, i.p.), ANI/NEO (20 mg/kg and 40 μg/kg, i.p.), 3-MA +ANI/NEO, or MLA+ANI/NEO. We administered 3-MA and MLA daily and ANI/NEO twice a day from the third to the seventh day. We then recorded body weight, DAI, colon length, histologic score, and INF-γ, TNF-α, IL-6, IL-22 expression in colonic tissue (by RT-PCR) as described above.

## Statistical analysis

Data are expressed as means ± SD. A repeated measurement analysis of variance (ANOVA) followed by LSD post *hoc test* was used for the analysis of body weight change and disease activity index. For other analysis, we compared the groups using one-way ANOVA followed by LSD *post hoc* test. We performed the statistical analyses using SPSS 21.0K (SPSS, Chicago, IL, USA) and considered P <0.05 as statistically significant.

## Results

### The ANI and NEO combination alleviates the symptoms and severity of DSS-induced colitis

C57BL/6 mice received 3% DSS for seven days, while control mice received tap water. The DSS group mice developed a severe illness characterized by sustained weight loss, bloody diarrhea, severe colon inflammation, hyperemia, ulceration, and bowel wall thickening. The ANI-treated and NEO-treated groups had higher body weights than the DSS group. However, the ANI/NEO-treated group had markedly higher body weight than the ANI-treated and NEO-treated groups (Fig 1A). As shown in Fig 1B and 1C, the ANI-treated and NEO-treated groups had lower DAIs and longer colons than the DSS group. Furthermore, the ANI/NEO-treated group had markedly lower DAI and longer colons than the ANI-treated and NEO-treated groups. Besides, the ANI/NEO treatment relatively alleviated inflammatory infiltration in the mucosa and submucosa of DSS-treated mice (Fig 1D).

### Dose-dependent effects of the ANI/NEO combination at 500:1 ratio on the symptoms and severity of DSS-induced colitis

To find the most efficient ANI/NEO dose, we administered three different doses (5 mg/kg and 10 μg/kg, 10 mg/kg and 20 μg/kg, 20 mg/kg and 40 μg/kg, i.p.)—all at a 500:1 ANI to NEO ratio—to mice twice a day from the third to the seventh day of DSS treatment. The two higher ANI/NEO doses significantly reduced body weight loss (Fig 2A), improved the DAI (Fig 2B), and enhanced colon length (Fig 2C and 2D). Moreover, these two ANI/NEO doses remarkably reduced colon inflammation, and the highest dose was more effective (Fig 2E). In light of these results, we used the highest dose in the subsequent experiments.

### The ANI/NEO combination increases autophagy and alleviates colon inflammation in mice with DSS-induced colitis

Since autophagy mediates the homeostasis of intestinal function and inflammatory cytokines expression, we quantified the autophagy-related proteins LC3, Beclin-1, P62, and proinflammatory cytokines. The ANI/NEO combination significantly increased the LC3-II/LC3-I ratio and Beclin-1 expression and decreased P62 levels in colons from DSS-treated mice (Fig 3A).

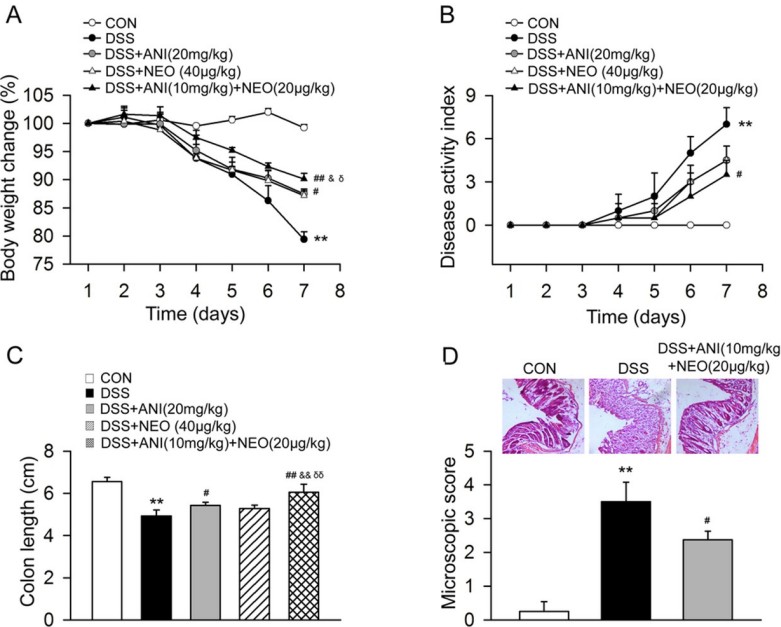

**Fig 1. Effects of ANI and NEO on the symptoms and colon inflammation in DSS-induced colitis mice.** C57BL/6 mice were received 3% DSS for 7 days, the control mice were given tap water. Animals were treated with vehicle, ANI (20 mg/kg, i.p.), NEO (40 μg/kg, i.p.) or ANI/NEO compound (10 mg/kg and 20 μg/kg, i.p.) twice a day from day 3 to day 7. (A) The body weight loss was tested in the mice. n = 3 per group. **p < 0.01 vs. control groups; #P<0.05 vs. DSS group; ##P<0.01 vs. DSS group; &P<0.05 vs. ANI group; Q̄P<0.05 vs. NEO group; (B) The disease activity index was tested in the mice. n = 4 per group. **p < 0.01 vs. control groups; #P<0.05 vs. DSS group; (C) The colon length was tested in the mice. n = 4 per group. **P<0.01 vs. control groups; #P<0.05 vs. DSS group; ##P<0.01 vs. DSS group. &&P<0.05 vs. ANI group; Q̄Q̄P<0.05 vs. NEO group; (D) The colon inflammation in the DSS-induced colitis mice. n = 4 per group. **P<0.01 vs. control groups; #P<0.05 vs. DSS group.

Besides, the protein level of autophagy related gene 5 (ATG5) was increased after DSS treatment and ANI/NEO combination promoted the elevation of ATG5 furtherly (S1 Fig). In addition, the ANI/NEO combination noticeably decreased the INF-γ, TNF-α, IL-6, and IL-22 mRNA levels and the TNF-α and INF-γ concentrations in colon tissue of DSS-treated mice (Fig 3B). As IL-1β is significantly elevated in mice with UC and plays a pivotal role in the pathogenesis of UC, we tested the concentration of IL-1β in colon tissues with ELISA. The result showed that ANI/NEO combination significantly decreased IL-1β from DSS-treated mice (S2 Fig). Furthermore the western blotting showed that ANI/NEO treatment inhibited DSS-induced expression of p-NF-κB p65 (S1 Fig). This finding indicates that inhibition of NF-κB partly mediates the anti-inflammatory activity of ANI/NEO.

## The ANI/NEO combination increases autophagy in LPS/DSS-treated Caco-2 cells

We primed Caco-2 cells with 10 ng/ml LPS for 1 h, then stimulated them with 3% DSS and 0, 10, 30, or 100 μM of ANI/NEO for 24 h. ANI/NEO dose-dependently enhanced LC3-II/LC3-I ratio and Beclin-1 levels and reduced P62 levels (Fig 4A). Since autophagosomes are basic functional units of autophagy, we next quantified autophagosomes in Caco-2 cells. LPS/DSS stimulation greatly increased the number of autophagosomes. Treatment with ANI/NEO (100 μM) enhanced the effect of LPS/DSS on autophagosomes (Fig 4B).

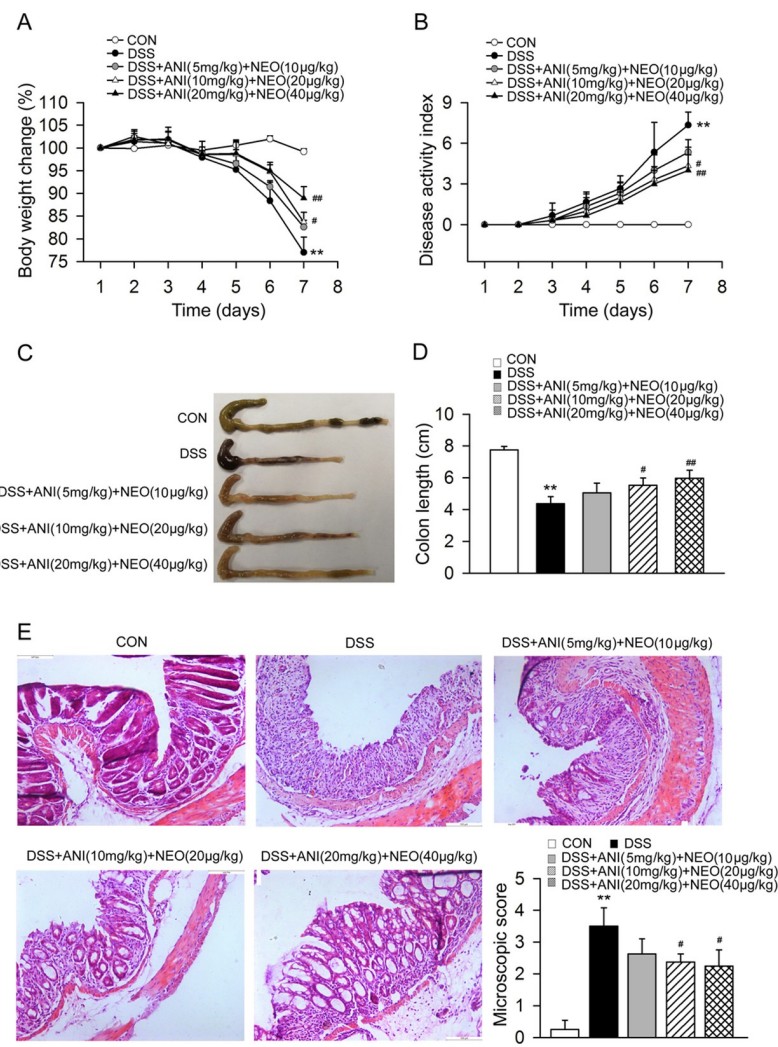

**Fig 2. Effects of different dose of ANI/NEO combination on the symptoms and colon inflammation in DSS-induced colitis mice.** C57BL/6 mice were received 3% DSS for 7 days, the control mice were given tap water. Animals were treated with vehicle or ANI/NEO compound at three different doses (5 mg/kg and 10 μg/kg, 10 mg/kg and 20 μg/kg, 20 mg/kg and 40 μg/kg, i.p.) twice a day from day 3 to day 7. (A) The body weight loss was tested in the mice. n = 3 or 4 per group. **P<0.01 vs. the control group; #P<0.05 vs. the DSS group, ##P<0.01 vs. the DSS group; (B) The disease activity index was tested in the mice. n = 6 per group. **P<0.01 vs. the control group; #P<0.05 vs. the DSS group, ##P<0.01 vs. the DSS group; (C-D) The colon length was tested in the mice. n = 4 per group. **P<0.01 vs. the control group; #P<0.05 vs. the DSS group, ##P<0.01 vs. the DSS group. (E) The colon inflammation was tested in the mice. n = 4 per group. **P<0.01 vs. the control group; #P<0.05 vs. the DSS group.

## Autophagy mediates the ANI/NEO-induced inflammatory cytokines expression inhibition in LPS/DSS-treated Caco-2 cells

Intestinal immune dysfunction-induced intestinal tissue inflammation plays an important role in the pathogenesis of UC. The inflammatory cytokines levels and proportions in the intestinal tissue are closely related to the pathogenesis of UC. LPS/DSS stimulation increased TNF-α, IL-1β, and IL-6 levels in Caco-2 cells. Meanwhile, preincubation with ANI/NEO significantly reduced this effect. Finally, blocking autophagy using ATG5 siRNA attenuated the inhibitory effect of ANI/NEO on the expression of TNF-α, IL-1β, and IL-6 (Fig 5A–5D).

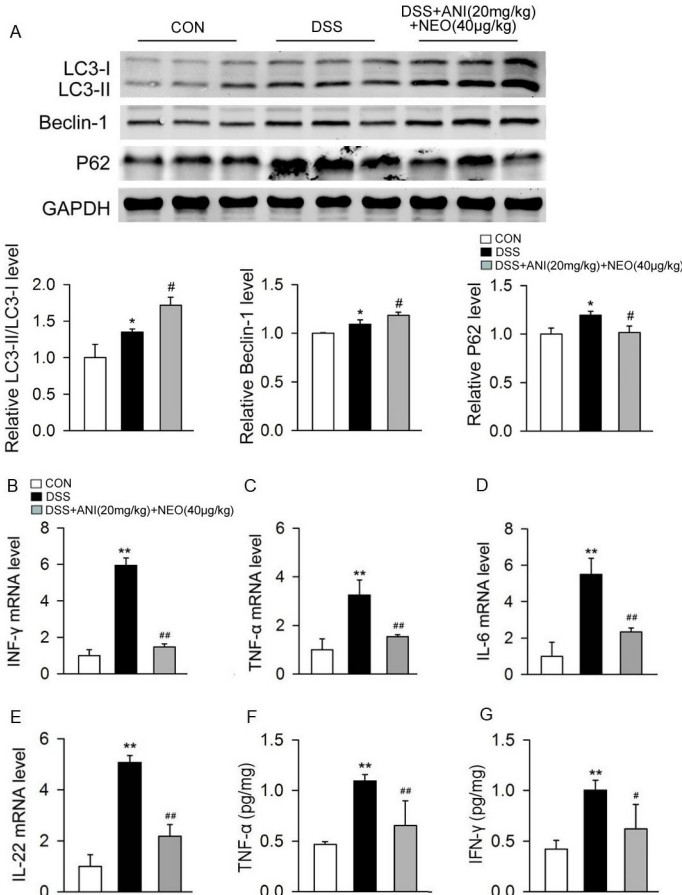

**Fig 3. Effects of ANI/NEO combination on autophagy and colonic inflammatory cytokines in DSS-induced colitis mice.** C57BL/6 mice were received 3% DSS for 7 days, the control mice were given tap water. Animals were treated with vehicle or ANI/NEO compound (20 mg/kg and 40 µg/kg, i.p.) twice a day from day 3 to day 7. The protein levels of LC3-II/LC3-I ratio, Beclin-1, P62, the concentration of INF-γ, TNF-α and the mRNA levels of INF-γ, TNF-α, IL-6, IL-22 in colonic tissue were detected. (A) Representative western blot and quantification data for LC3-II/LC3-I, Beclin-1 and P62 level in the intestinal tissues. n = 3 per group. *P<0.05 vs. the control group; #P<0.05 vs. the DSS group. (B-E) The INF-γ, TNF-α, IL-6, and IL-22 mRNA levels were detected in the mice. n = 3 per group. **P<0.01 vs. the control group; ##P<0.01 vs. the DSS group. (F-G) The TNF-α and INF-γ protein levels were detected in the mice. n = 4 per group. **P<0.01 vs. the control group; #P<0.05 vs. the DSS group, ##P<0.01 vs. the DSS group.

## Blocking autophagy attenuates the protective effect of ANI/NEO on DSS-induced colitis and inflammation

Administering the autophagy blocker 3-MA (10mg/kg, i.p.) to mice with DSS-induced colitis attenuated the beneficial effects of ANI/NEO on weight loss, DAI, and colon length (Fig 6A–6D). We thus assessed the effect of 3-MA on intestinal inflammation. The ANI/NEO-treated group had significantly lower proinflammatory factors levels (TNF-α, IL-6, IFN-γ, IL-22) in colon tissue than the DSS-treated group. Besides, 3-MA increased the proinflammatory factors levels (TNF-α, IL-6, IFN-γ, IL-22) in ANI/NEO-treated mice with DSS-induced colitis. Finally, 3-MA partially weakened the inhibitory effect of ANI/NEO on inflammation (Fig 7). These results indicate that autophagy mediates the protective effect of ANI/NEO on DSS-induced intestinal inflammation.

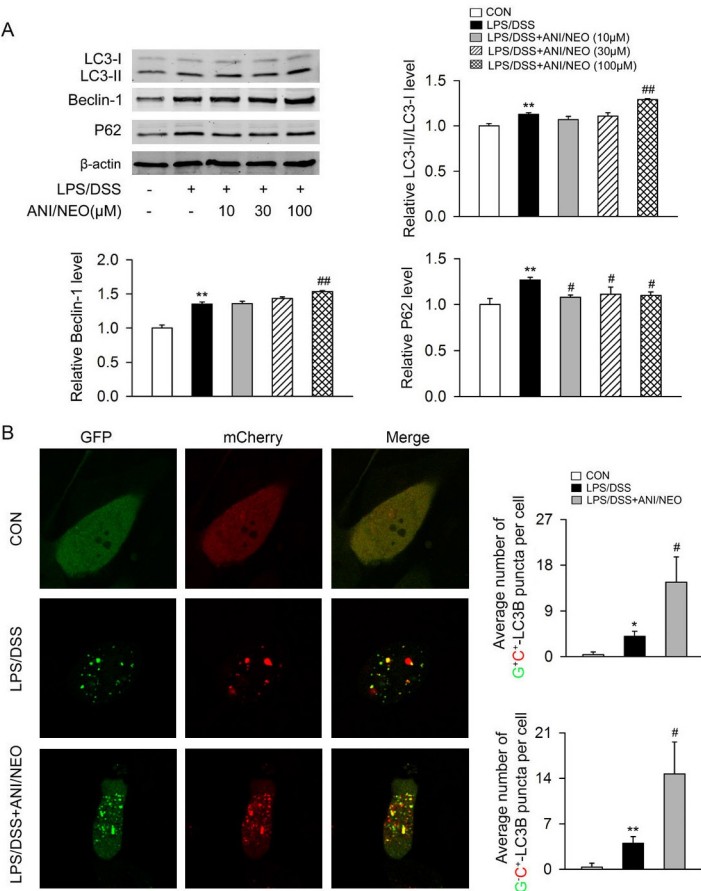

**Fig 4. Effects of ANI/NEO combination on autophagy in LPS/DSS simulated Caco-2 cells.** (A) Caco-2 cells were primed with 10 ng/ml LPS for 1 h, and then stimulated with 3% DSS in the presence or absence of 10, 30, 100 μM ANI/NEO compound for 24 h. The LC3-II/LC3-I ratio, Beclin-1 and P62 in Caco-2 cells were analyzed. n = 3 per group. **P<0.01 vs. the control group; ##P<0.01 vs. the DSS group. (B) Caco-2 cells were transfected with tandem fluorescent AdPlus-mCherry-GFP-LC3B plasmid for 8 h and then cultured in fresh DMEM supplemented with 10% (vol/vol) FBS for another 48 h. After primed with 10 ng/ml LPS for 1 h, cells were stimulated with 3% DSS in the presence or absence of 100 μM ANI/NEO compound for 24 h. Cellular autophagosomes (G+C+) and autolysosomes (G−C+) were detected. ANI/NEO significantly increased the number of autophagosomes (G+C+) and autolysosomes (G−C+). n = 3 per group. *P<0.05 vs. the control group, **P<0.01 vs. the control group; #P<0.05 vs. the DSS group.

## Blocking α7nAChR partly inhibits the protective effect of ANI/NEO on DSS-treated mice

MLA (10 mg/kg, i.p.), a selective inhibitor of α7nAChR, abolished the inhibitory effects of ANI/NEO on weight loss, DAI, and colon length reduction in mice with DSS-induced colitis (Fig 8). Then we examined the effect of MLA on proinflammatory cytokines in colons from DSS treated mice. The ANI/NEO combination significantly inhibited the mRNA levels of TNF-α, IL-6, IFN-γ, and IL-22 in colon tissue induced by DSS. MLA partly abolished the inhibitory effect of ANI/NEO combination on these proinflammatory cytokines (S3 Fig). These results suggest that ANI/NEO protects mice from DSS-induced colitis by activating α7nAChR.

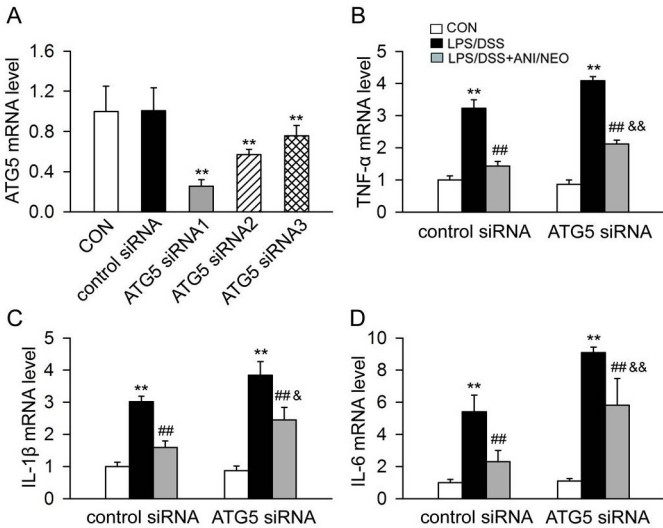

**Fig 5. Autophagy mediates the inhibition of ANI/NEO combination on the expression of inflammatory cytokines in Caco-2 cells challenged with LPS/DSS.** (A) The quantitative expression of ATG5 mRNA in Caco-2 cells after transfection with control siRNA or ATG5 siRNA. n = 3 per group. **P<0.01 vs. the control siRNA. (B-D) After transfection with control siRNA or ATG5 siRNA for 24h, Caco-2 cells were primed with 10ng/ml LPS for 1 h, and then stimulated with 3% DSS in the presence or absence of 100 μM ANI/NEO compound for 24 h. The mRNA levels of TNF-α, IL-1β and IL-6 were analyzed with RT-PCR. n = 3 per group. **P<0.01 vs. the corresponding control group; ##P<0.01 vs. the corresponding LPS/DSS group; &P<0.05 vs. the control siRNA LPS/DSS+ANI/NEO group, &&P<0.01 vs. the control siRNA LPS/DSS +ANI/NEO group.

## Discussion

This study is the first to show that combining ANI and NEO markedly alleviated DSS-induced colitis symptoms. This treatment reduced weight loss, DAI, and colon shortening, remarkably improving histopathology scores. Additionally, we found that the best ANI to NEO ratio was 500:1. Besides elevating the LC3-II/LC3-I ratio, ATG5 and Beclin-1 levels and decreasing P62 level, the ANI/NEO combination decreased the expression of several inflammatory cytokines (INF-γ, TNF-α, IL-6, and IL-22) in colon tissue from mice with DSS-induced colitis. We obtained similar results in LPS/DSS-stimulated Caco-2 cells. The ANI/NEO pre-incubation increased the LC3-II/LC3-I ratio and Beclin-1 level, decreased P62 level, promoted autophagy flux, and reduced TNF-α, IL-1β, and IL-6 mRNA levels. These *in vivo* and *in vitro* experiment results suggest that ANI/NEO enhances autophagy and inhibits the inflammatory response.

Recent studies have reported that ANI regulates the inflammatory response in inflammatory and autoimmune diseases [18–20]. Our previous works showed that ANI markedly reduced TNF-α and IL-1β levels in rats with LPS-induced septic shock [21]. Next, we found that NEO, an acetylcholinesterase inhibitor, greatly potentiated the anti-inflammatory effect of ANI. The combination treatment displayed superior therapeutic efficiency in septic shock, acute lethal crush syndrome, and collagen-induced arthritis mouse models [11, 12, 22]. In line with those studies, the present work showed that the ANI/NEO combination significantly reduced body weight loss, improved the DAI, enhanced colon length, and alleviated colon inflammation in a DSS-induced colitis mouse model. Furthermore, combining ANI and NEO had a more potent protective effect against DSS-induced colitis than ANI or NEO alone.

During UC development, the intestinal lamina propria macrophages, derived from circulating monocytes, increase and excessively produce inflammatory cytokines, stimulating neutrophil infiltration into the colon and damaging intestinal epithelial tissue [1]. Yan found that the

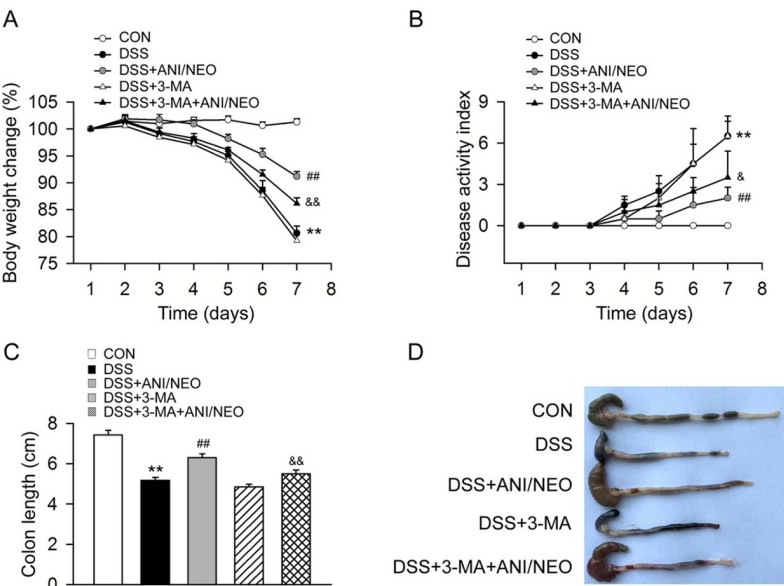

**Fig 6. 3-MA attenuates the protective effect of ANI/NEO combination on DSS-induced colitis in mice.** C57BL/6 mice were received 3% DSS for 7 days, the control mice were given tap water. The DSS mice were treated with vehicle, 3-methyladenine (3-MA, 10 mg/kg, i.p.), ANI/NEO compound (20 mg/kg and 40 μg/kg, i.p.) or 3-MA+ ANI/NEO compound. 3-MA was given daily and ANI/NEO compound twice a day from day 3 to day 7. (A) The body weight loss was tested in the mice. n = 4 per group. **$P<0.01$ vs. the control group; ##$P<0.01$ vs. the DSS group. &&$P<0.01$ vs. the DSS+ANI/NEO group; (B) The disease activity index was tested in the mice. n = 4 per group. **$P<0.01$ vs. the control group; ##$P<0.01$ vs. the DSS group. &$P<0.05$ vs. the DSS+ANI/NEO group; (C-D) The colon length was tested in the mice. n = 4 per group. **$P<0.01$ vs. the control group; ##$P<0.01$ vs. the DSS group. &&$P<0.01$ vs. the DSS+ANI/NEO group.

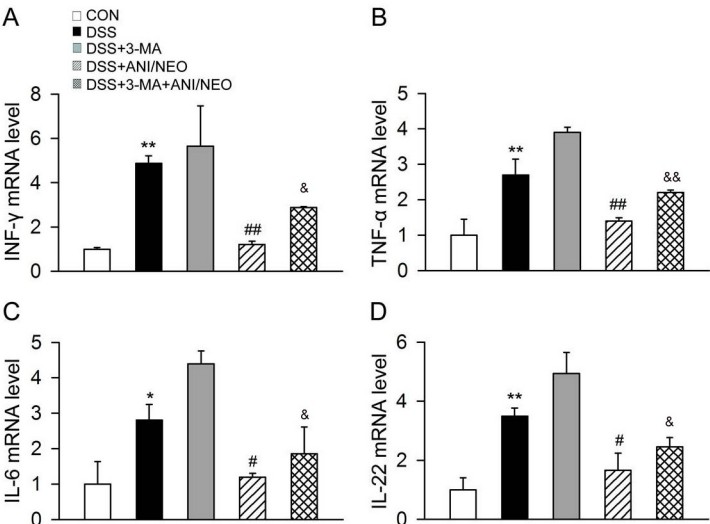

**Fig 7. 3-MA attenuates the inhibitory effect of ANI/NEO combination on inflammatory cytokines in colon from DSS-induced colitis mice.** C57BL/6 mice were received 3% DSS for 7 days, the control mice were given tap water. The DSS mice were treated with vehicle, 3-MA (10 mg/kg, i.p.), ANI/NEO compound (20 mg/kg and 40 μg/kg, i.p.) or 3-MA+ ANI/NEO compound. 3-MA was given daily and ANI/NEO compound twice a day from day 3 to day 7. The expression of INF-γ, TNF-α, IL-6 and IL-22 in colonic tissue were detected with RT-PCR. n = 3 per group. *$P<0.05$ vs. the control group, **$P<0.01$ vs. the control group; #$P<0.05$ vs. the DSS group, ##$P<0.01$ vs. the DSS group; &$P<0.05$ vs. the DSS+ANI/NEO group, &&$P<0.01$ vs. the DSS+ANI/NEO group.

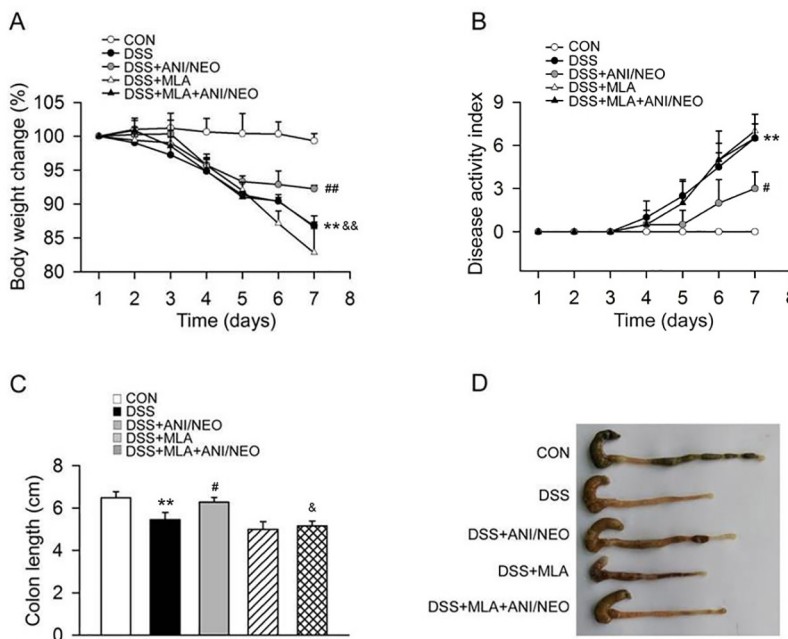

**Fig 8. Methyllycaconitine (MLA) significantly inhibits the protective effect of ANI/NEO combination on DSS-induced colitis.** C57BL/6 mice were received 3% DSS for 7 days, the control mice were given tap water. The DSS mice were treated with vehicle, MLA (10 mg/kg, i.p.), ANI/NEO compound (20 mg/kg and 40 μg/kg, i.p.) or MLA+ ANI/NEO compound. MLA was given daily and ANI/NEO compound twice a day from day 3 to day 7. (A) The body weight loss was tested in the mice. n = 3 per group. **$P<0.01$ vs. the control group; ##$P<0.01$ vs. the DSS group; &&$P<0.01$ vs. the DSS+ANI/NEO group. (B) The disease activity index was tested in the mice. n = 4 per group. **$P<0.01$ vs. the control group; #$P<0.05$ vs. the DSS group. (C-D) The colon length was tested in the mice. n = 4 per group. **$P<0.01$ vs. the control group; #$P<0.05$ vs. the DSS group; &$P<0.05$ vs. the DSS+ANI/NEO group.

artemisinin analog SM934 alleviated DSS-induced UC in mice by inhibiting neutrophils and macrophages infiltration and IL-1β, IL-6, and TNF-α production [23]. Moreover, cytokines are potential targets for new IBD treatments [24]. Anti-TNF-α agents like infliximab ameliorate DAI and histological score in DSS-induced colitis mouse models and induce clinical remission and mucosal healing in UC patients [25]. In this work, we found that administering the ANI/NEO combination at 20 mg/kg and 40 μg/kg significantly decreased the INF-γ, TNF-α, IL-6, and IL-22 mRNA levels and TNF-α, INF-γ, and IL-1β concentrations in colon tissues of DSS-treated mice. Besides, in LPS/DSS-stimulated Caco-2 cells, ANI/NEO (100 μM) also reduced TNF-α, IL-1β, and IL-6 mRNA levels. These results suggest that ANI/NEO alleviates DSS-induced colitis symptoms by inhibiting the inflammatory reaction. NF-kB is a pivotal mediator of the immunological reaction in IBD and activation of NF-kB in the intestine promotes the production of proinflammatory cytokines. In our study, we found that ANI/NEO treatment significantly inhibited the phosphorylation of NF-kB p65 in colons from DSS-treated mice, which illustrates the repression of ANI/NEO on inflammatory reaction could be attributed to NF-kB inhibition.

Previous studies and our works demonstrated that autophagy plays a crucial role in UC by modulating the inflammatory response [26]. For example an intestinal epithelial cell-specific deletion of ATG16 severely exacerbated colitis pathology, elevated proinflammatory cytokine secretion, and increased intestinal epithelial cell apoptosis. Thus, in the epithelium, autophagy prevents intestinal damage by regulating inflammation and apoptosis [27]. Furthermore, our previous studies demonstrated that inducing autophagy by activating the cannabinoid receptor

2 inhibited NLRP3 inflammasome activation in colitis and EAE mouse models [9, 28]. Here, we revealed that the ANI/NEO combination enhanced autophagy both in the colon of mice with DSS-induced colitis and LPS/DSS-stimulated Caco-2 cells. In addition, blocking autophagy using ATG5 siRNA in Caco-2 cells attenuated the effect of ANI/NEO on TNF-α, IL-1β, and IL-6. Likewise, 3-MA, an autophagy inhibitor, notably attenuated the protective effect of ANI/NEO on DSS-induced colitis and its inhibitory effect on cytokines production. Taken together, these results suggest that inducing autophagy at least partly contributed to the ANI/NEO-mediated inflammatory response suppression in Caco-2 cells and the DSS-induced colitis mouse model.

A clinical investigation reported that smoking inhibited the progression and severity of UC, while stopping smoking increased the recurrence rate. This occurs because nicotine activates α7nAChR, producing an anti-inflammatory effect [13, 29, 30]. Nicotine can also activate α7nAChR on CD4T cells in the colonic mucosa and inhibit IL-6, STAT3, and TNF-α production, preventing UC [31]. In addition, Alsharari, *et al.* demonstrated that α7nAChR knockout mice displayed markedly elevated colitis severity and TNF-α levels. In contrast, pretreating mice with selective α7nAChR agonists, PHA-543613 and PNU120596, alleviated the colitis symptoms [32]. Similarly, selective α7nAChR agonists, PNU282987 and Enenicline, reduced enteritis immune inflammation and UC [33, 34]. In this work, we found that MLA, a selective α7nAChR inhibitor, noticeably inhibited the protective effect of ANI/NEO on DSS-induced colitis and intestinal proinflammatory cytokines indicating that α7nAChR mediates the therapeutic effect of ANI/NEO. Interestingly, some studies reported that AR-R17779 and GSK1345038A, two selective α7nAChR agonists, aggravated UC and increased the levels of colonic proinflammatory cytokines [35]. Several experimental factors could explain this discrepancy, such as α7nAChR agonist selectivity, dosage, administration mode, disease severity, and disease model variation.

The α7nAChR is highly expressed in the hypothalamus and hippocampus as well as in glial cells [36, 37]. Activation of α7nAChR can inhibit the release of pro-inflammatory cytokines such as IL-6, TNF-α, and IL-1β. Moreover, the expression of α7nAChR is also found on immune cells, which exhibits cytokine inhibitory effect through the cholinergic anti-inflammatory pathway [38, 39]. In our previous study we found that autophagy mediated the anti-inflammatory effect of α7nAChR on EAE [10]. Furthermore, blocking α7nAChR with MLA decreased the phosphorylation of Akt and abolished the induction of autophagy by dexmedetomidine, leading to increased inflammation and cell death in CLP-induced sepsis [40]. Primary chondrocytes from α7nAChR gene knockout mice showed lower autophagy level compared to cells from wild type mice and α7nAChR deletion increased the level of p-mTOR after MIA treatment in primary chondrocytes as well as in tissues from OA patients [41]. These findings showed that α7nAChR may promote autophagy through Akt and mTOR signal pathway. In this work, blocking of α7nAChR by MLA in ANI/NEO treated colitis mice, the levels of LC3-II/LC3-I ratio and Beclin-1 were decreased and the P62 was increased, suggesting that α7nAChR mediates the role of ANI/NEO on promoting autophagy in DSS-induced mice.

## Conclusions

In conclusion, our study demonstrated that combining ANI and NEO at a 500:1 ratio produced therapeutic effects on a DSS-induced colitis mouse model, at least partly through autophagy induction and inflammation inhibition. Furthermore, the protective effect of ANI/NEO is related to α7nAChR. This study may provide a new therapeutic strategy for UC, but the path from bench to bedside is long, and side-effects, preparation stability, pharmacokinetics, efficacy, and toxicology studies need to be conducted.

## Supporting information

**S1 Fig. Effects of ANI/NEO combination on ATG5 and NF-κB in DSS-induced colitis mice.** C57BL/6 mice were received 3% DSS for 7 days, the control mice were given tap water. Animals were treated with vehicle or ANI/NEO compound (20 mg/kg and 40 μg/kg, i.p.) twice a day from day 3 to day 7. The protein level of ATG5 and the phosphorylation level of NF-κB in colonic tissue were detected. Representative western blot and quantification data for ATG5 and p-NF-κB p65 in the intestinal tissues. n = 3 per group. *P<0.05 vs. the control group, **P<0.01 vs. the control group; #P<0.05 vs. the DSS group, ##P<0.01 vs. the DSS group.
(TIF)

**S2 Fig. Effects of ANI/NEO combination on IL-1β in DSS-induced colitis mice.** C57BL/6 mice were received 3% DSS for 7 days, the control mice were given tap water. Animals were treated with vehicle or ANI/NEO compound (20 mg/kg and 40 μg/kg, i.p.) twice a day from day 3 to day 7. The concentration of IL-1β in colonic tissue were detected. n = 4 per group. **P<0.01 vs. the control group; ##P<0.01 vs. the DSS group.
(TIF)

**S3 Fig. MLA attenuates the inhibitory effect of ANI/NEO combination on inflammatory cytokines in colon from DSS-induced colitis mice.** C57BL/6 mice were received 3% DSS for 7 days, the control mice were given tap water. The DSS mice were treated with vehicle, MLA (10 mg/kg, i.p.), ANI/NEO compound (20 mg/kg and 40 μg/kg, i.p.) or MLA+ ANI/NEO compound. MLA was given daily and ANI/NEO compound twice a day from day 3 to day 7. The expression of INF-γ, TNF-α, IL-6 and IL-22 in colonic tissue were detected with RT-PCR. n = 3 or 4 per group. **P<0.01 vs. the control group; ##P<0.01 vs. the DSS group.
(TIF)

**S4 Fig. The effect of ANI/NEO combination on autophagy in Caco-2 cells challenged with LPS/DSS after knocking down ATG5.** After transfection with control siRNA or ATG5 siRNA for 24h, Caco-2 cells were primed with 10 ng/ml LPS for 1 h, and then stimulated with 3% DSS in the presence or absence of 100 μM ANI/NEO compound for 24 h. (A-B) The LC3-II/LC3-I ratio in Caco-2 cells were analyzed. n = 3 per group. **P<0.01 vs. the Control siRNA LPS/DSS group; ##P<0.01 vs. the Control siRNA LPS/DSS+ANI/NEO group.
(TIF)

**S5 Fig. α7nAChR mediates the protective effect and autophagy of ANI/NEO combination on DSS-induced colitis in mice.** WT and α7nAChR KO mice were received 3% DSS for 7 days and the DSS mice were treated with vehicle or ANI/NEO compound (20 mg/kg and 40 μg/kg, i.p.) twice a day from day 3 to day 7. (A) The body weight loss was tested in the mice. n = 4 per group. #P<0.05 vs. the WT DSS group. (B) The disease activity index was tested in the mice. n = 4 per group. #P<0.05 vs. the WT DSS group. (C-D) The colon length was tested in the mice. n = 4 per group. **P<0.01 vs. the WT DSS group. (E-F) The LC3-II/LC3-I ratio in colonic tissues were analyzed. n = 3 per group. **P<0.01 vs. the WT DSS group; ##P<0.01 vs. the WT DSS+ANI/NEO group.
(TIF)

**S1 Data.**
(ZIP)

**S2 Data.**
(RAR)

**S1 Raw images.**
(PDF)

**S2 Raw images.**
(PPTX)

## Acknowledgments

We would like to thank all the authors who contributed papers to this collection. We thank Home for Researchers editorial team (www.home-for-researchers.com) for language editing service. We would also like to thank the PLOS ONE staff for their valuable support.

## Author Contributions

**Conceptualization:** Chong Liu.

**Data curation:** Jingjing Zhang, Guangyu Zhang, Ting Han.

**Investigation:** Ping Ke, Chong Liu.

**Methodology:** Mengzhen Liu, Danni Zhu, Hui Yan, Jingjing Zhang, Ni Kong, Guangyu Zhang, Qin Xu, Ping Ke.

**Resources:** Zhiwei Dong, Ping Ke, Chong Liu.

**Supervision:** Chong Liu.

**Validation:** Zhiwei Dong, Chong Liu.

**Visualization:** Ping Ke.

**Writing – original draft:** Ping Ke.

**Writing – review & editing:** Chong Liu.

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
