## [Decision Letter · Decision Letter 0]

28 Nov 2022

PONE-D-22-21418Combined Administration of Anisodamine and Neostigmine Alleviated Dextran Sulfate Sodium-induced Colitis by Inducing Autophagy and Inhibiting InflammationPLOS ONE

Dear Dr. Ke,

Thank you for submitting your manuscript to PLOS ONE. First of all we are sorry for the delay in the review process. After careful consideration, we feel that it has merit but does not fully meet PLOS ONE’s publication criteria as it currently stands. Therefore, we invite you to submit a revised version of the manuscript that addresses the points raised during the review process.

The main issues relate to the lack of rationale for certain studies and need for more in depth look at the mechanism,  highlighting the novelty of the presented study as well as a point by point response to reviewers comments would be important.

We look forward to receiving your revised manuscript.

Kind regards,

Pradeep Dudeja

Academic Editor

PLOS ONE

Journal Requirements:

2. To comply with PLOS ONE submissions requirements, in your Methods section, please provide additional information on the animal research and ensure you have included details on (1) methods of sacrifice, (2) methods of anesthesia and/or analgesia, and (3) efforts to alleviate suffering.

“This work was supported by grants from the National Natural Science Foundation of China (81903630 to PK) and Shanghai Biomedical Science and Technology Support Program (20S190280 to CL) in study design, data collection and analysis. It was also supported by National Natural Science Foundation of China (81871596 CL and 81800473 to ZWD) in decision to publish and preparation of the manuscript.”

Reviewers' comments:

Reviewer's Responses to Questions

**Comments to the Author**

1. Is the manuscript technically sound, and do the data support the conclusions?

Reviewer #2: Yes

Reviewer #3: Yes

2. Has the statistical analysis been performed appropriately and rigorously? 

Reviewer #2: Yes

Reviewer #3: Yes

3. Have the authors made all data underlying the findings in their manuscript fully available?

Reviewer #2: Yes

Reviewer #3: Yes

4. Is the manuscript presented in an intelligible fashion and written in standard English?

Reviewer #2: Yes

Reviewer #3: Yes

5. Review Comments to the Author

Reviewer #2: The study by Ping Ke et al. entitled “Combined Administration of Anisodamine and Neostigmine Alleviated Dextran Sulfate Sodium-induced Colitis by Inducing Autophagy and Inhibiting Inflammation” examined the beneficial effect of anisodamine and neostigmine on intestinal inflammation. However, there are some concerns, which can be addressed to strengthen the manuscript.

1)The authors did not provide the rationale for using ANI and NEO in intestinal inflammation. Please add a details explanation to the manuscript.

2)To provide more mechanistic insight, authors can also examine the phosphorylation levels of STAT3 and NFkB in the same region, as these are the central mediators of inflammation.

3) IL-1B is the main cytokine, which is known to be upregulated up to thousand-fold in DSS treated mice. It would be important to study the main cytokine which exerts the effect after DSS treatment.

4)It would be also great to examine the cytokines levels in mice samples treated with MLA as shown in fig 8.

Reviewer #3: This manuscript elucidates the therapeutic effect and mechanisms underlying the role of anisodamine (ANI)/neostigmine (NEO) combination in dextran sulfate sodium (DSS)-induced colitis and lipopolysaccharide/DSS-stimulated intestinal epithelial Caco2 cells. The authors showed that treating mice with ANI and NEO at a ratio of 500:1 alleviated the DSS-induced colitis symptoms, reduced body weight loss, improved the disease activity index, enhanced colon length, and alleviated colon inflammation. In addition to decreasing the expression of several inflammatory cytokines (IFN-γ, TNF-α, IL-6, IL-22 and IL1�) in the colon of DSS-induced colitis mice and LPS/DSS-stimulated Caco2 cells, ANI/NEO combination also elevated autophagy markers, LC3-II/LC3-I ratio and Beclin-1 levels and decreased P62 levels. Furthermore, autophagy blocker, 3-methyladenine and the α7nAChR antagonist, methyllycaconitine blunted the protective effects of ANI/NEO in DSS-induced colitis mice. Overall, both the in vivo and in vitro studies indicated that ANI/NEO effects occurred via an increase in autophagy and inhibition of the inflammatory response. The experiments are carefully performed, and the author's conclusions are well supported by the presented data. However, the study is not entirely novel and appears redundant. In this regard, previous studies have shown that selective α7nAChR agonists like ANI/NEO (activate α7nAChR) reduced intestinal inflammation in DSS or TNBS induced colitis mice (Refs. 32, 33, 34) and suppressed neuroinflammation in experimental autoimmune encephalomyelitis (EAE) mice via an autophagy-dependent mechanism (Ref. 10). The authors should consider the following suggestions:

1. The authors should highlight what is novel in the current study compared to previous studies (Refs. 10, 32, 33 and 34).

2. The authors showed opposing effects of DSS and ANI/NEO on various pro-inflammatory cytokines, while both DSS and ANI/NEO showed similar induction in autophagy markers (LC3-II/LC3I ratio and Beclin 1 levels). It is suggested that ANI/NEO mediates its protective effects by inducing autophagy. However, DSS was also shown to increase LC3-II/LC3I ratio and Beclin 1 levels. Is DSS also protective? How do the authors decipher this aspect?

3. It will be helpful to assess the protein levels of other autophagy markers such as ATG5 and ATG7 in the colon of DSS colitis mice. This will support the in vitro ATG5 siRNA studies in Caco2 cells.

6. PLOS authors have the option to publish the peer review history of their article (what does this mean?). If published, this will include your full peer review and any attached files.

Reviewer #1: No

Reviewer #2: No

Reviewer #3: **Yes: **Seema Saksena

---

## [Author Response · Author response to Decision Letter 0]

2 Jun 2023

ITEMIZED RESPOSES TO REVIEWERS’ COMMENTS

To Reviewer 2:

1.The authors did not provide the rationale for using ANI and NEO in intestinal inflammation. Please add a details explanation to the manuscript.

Re: Our groups have studied the therapeutic effects of anisodamine and neostigmine for many years, and found out that the anti-inflammatory effect is best when the ratio of anisodamine and neostigmine is at 500:1 in a variety of animal models (septic shock, acute lethal crush syndrome and collagen-induced arthritis, etc.) (doi: 10.1038/aps.2012.26, doi: 10.1038/srep37709, doi: 10.1111/cns.12213). Therefore, we directly used anisodamine and neostigmine combination at this ratio in DSS-induced colitis. From the pharmacodynamic indicators of colitis, the anti-inflammatory effect of this combination is very effective, which is consistent with our previous research. 

2.To provide more mechanistic insight, authors can also examine the phosphorylation levels of STAT3 and NFkB in the same region, as these are the central mediators of inflammation.

Re: Thank you for your professional suggestion. The phosphorylation level of NF-kB p65 in colon tissue was examined by western blot (S1 figure, Line 233-236, Line 315-320).

3. IL-1B is the main cytokine, which is known to be upregulated up to thousand-fold in DSS treated mice. It would be important to study the main cytokine which exerts the effect after DSS treatment.

Re: Thank you for your professional suggestion. The levle of IL-1β in colon tissue was detected by ELISA in S2 figure (Line 230-233). 

4.It would be also great to examine the cytokines levels in mice samples treated with MLA as shown in fig 8.

Re: Thank you for your professional suggestion. The mRNA levels of INF-γ, TNF-α, IL-6, IL-22 in colon tissue were assessed by RT-PCR (S3 figure, Line 270-274). 

To Reviewer 3:

1. The authors should highlight what is novel in the current study compared to previous studies (Refs. 10, 32, 33 and 34).

Re: Our innovations are as follows: (1) ANI/NEO combination provided protective effect on DSS-induced colitis; (2) The target of ANI/NEO combination to protect colitis is α7nAChR, indicating that it is an a7 receptor agonist which is not reported previously; (3) After activating α7nAChR, ANI/NEO combination inhibited inflammation by promoting the autophagy of intestinal epithelial cells induced by DSS.

2. The authors showed opposing effects of DSS and ANI/NEO on various pro-inflammatory cytokines, while both DSS and ANI/NEO showed similar induction in autophagy markers (LC3-II/LC3I ratio and Beclin 1 levels). It is suggested that ANI/NEO mediates its protective effects by inducing autophagy. However, DSS was also shown to increase LC3-II/LC3I ratio and Beclin 1 levels. Is DSS also protective? How do the authors decipher this aspect?

Re: Autophagy is the main intracellular degradation mechanism, through which substances are transported to and degraded in lysosomes. As a dynamic circulatory system, autophagy maintains cell renovation and homeostasis. But excessive autophagy will destroy cell homeostasis and cuase pathological conditions (doi: 10.1080/15548627.2020.1847462, doi: https://doi.org/10.1016/j.cell.2011.10.026). The initial augmented autophagy induced by DSS is a compensatory reaction of cells, which is helpful to the maintenance of cell homeostasis ( doi: 10.1080/15548627.2020.1847460). However, excessive autophagy caused by continuous stimulation will raise cell apoptosis and aggravate colitis (doi: 10.3389/fphar.2021.697360). The autophagy enhancement after ANI/NEO treatment is only an appearance. Whether it is beneficial or harmful to cells depends on inflammation, tissue damage and other indicators. Altogether the effect of enhanced autophagy on diseases requires further detection of cell function and disease indicators.

3. It will be helpful to assess the protein levels of other autophagy markers such as ATG5 and ATG7 in the colon of DSS colitis mice. This will support the in vitro ATG5 siRNA studies in Caco2 cells.

Re: Thank you for your professional suggestion. The protein level of ATG5 in colon tissue was assessed by western blot and was showed in S1 figure (Line 225-227).

---

## [Decision Letter · Decision Letter 1]

26 Jun 2023

PONE-D-22-21418R1Combined Administration of Anisodamine and Neostigmine Alleviated Dextran Sulfate Sodium-induced Colitis by Inducing Autophagy and Inhibiting InflammationPLOS ONE

Dear Dr. Ke,

Thank you for submitting your manuscript to PLOS ONE. After careful consideration, we feel that it has merit but does not fully meet PLOS ONE’s publication criteria as it currently stands. Therefore, we invite you to submit a revised version of the manuscript that addresses the points raised during the review process.

While, the manuscript has significantly improved, some concerns remain which relate to the role of autophagy and the role of α7nAChR activation in promoting autophagy in the study. Please respond to the remaining concerns of the reviewer # 4.

We look forward to receiving your revised manuscript.

Kind regards,

Pradeep Dudeja

Academic Editor

PLOS ONE

Journal Requirements:

Reviewers' comments:

Reviewer's Responses to Questions

**Comments to the Author**

1. If the authors have adequately addressed your comments raised in a previous round of review and you feel that this manuscript is now acceptable for publication, you may indicate that here to bypass the “Comments to the Author” section, enter your conflict of interest statement in the “Confidential to Editor” section, and submit your "Accept" recommendation.

Reviewer #3: All comments have been addressed

Reviewer #4: (No Response)

2. Is the manuscript technically sound, and do the data support the conclusions?

Reviewer #3: Yes

Reviewer #4: Partly

3. Has the statistical analysis been performed appropriately and rigorously? 

Reviewer #3: Yes

Reviewer #4: Yes

4. Have the authors made all data underlying the findings in their manuscript fully available?

Reviewer #3: Yes

Reviewer #4: Yes

5. Is the manuscript presented in an intelligible fashion and written in standard English?

Reviewer #3: Yes

Reviewer #4: Yes

6. Review Comments to the Author

Reviewer #3: (No Response)

Reviewer #4: • Though authors have convincingly showed the protective effect of ANI/NEO combination on DSS-induced colitis, the role of autophagy and the role of α7nAChR activation in promoting autophagy is not clear. The relation between α7nAChR and autophagy is also not established. Thus, it is unclear if the protective effect of ANI/NEO is related to α7nAChR or autophagy, or both? Here are few things that may clarify the overall conclusions of this work.

• Fig. 4: In LPS/ ANI/NEO treated Caco-2 cells, LC3I and II can be blotted in the presence (and absence) of Bafilomycin A to demonstrate the effect of ANI/NEO on autophagic flux.

• As authors stated that ANI and NEO combination protects mice from DSS colitis by activating α7nAChR, is there a way to show that ANI/NEO actually activated α7nAChR or MLA inhibited α7nAChR in the mouse model?

• Does blocking of α7nAChR affects autophagy in mouse colon?

• It will be beneficial to add some information on α7nAChR in the discussion on what type of cells express α7nAChR and its physiological relevance and function, and possible mechanisms underlying its regulation by autophagy.

7. PLOS authors have the option to publish the peer review history of their article (what does this mean?). If published, this will include your full peer review and any attached files.

Reviewer #3: No

Reviewer #4: No

---

## [Author Response · Author response to Decision Letter 1]

8 Aug 2023

To Reviewer 4:

1.Fig. 4: In LPS/ ANI/NEO treated Caco-2 cells, LC3I and II can be blotted in the presence (and absence) of Bafilomycin A to demonstrate the effect of ANI/NEO on autophagic flux.

Re: Thanks for your suggestions! We blocked autophagy by ATG5 siRNA before LPS/DSS or LPS/DSS+ANI/NEO and then detected the expression of LC3 using WB. We found that LPS/DSS+ANI/NEO could significantly increase the LC3II/LC3I in the control siRNA group, whereas in ATG5siRNA group, LPS/DSS+ANI/NEO couldn’t increase the expression of LC3Ⅱ/LC3Ⅰ ratio (shown as Figure S4A-B).

2.As authors stated that ANI and NEO combination protects mice from DSS colitis by activating α7nAChR, is there a way to show that ANI/NEO actually activated α7nAChR or MLA inhibited α7nAChR in the mouse model?

Re: To explicit the role of α7nAChR in DSS-induced colitis, we compared the therapeutic effect of ANI/NEO on colitis between α7nAChR gene knockout and wild type mice. The findings showed that α7nAChR KO partly abolished the beneficial effects of ANI/NEO on weight loss, DAI, and colon length, which suggests that ANI/NEO at least partially activated α7nAChR (Figure S5A-D). 

MLA is a selective antagonist of α7nAChR and has been commonly used to confirm the role of α7nAChR in various disease models (doi: 10.1124/pharmrev.120.000097, doi: 10.1186/s12974-021-02341-6, doi: 10.1161/STROKEAHA.111.639989, doi: 10.1016/j.biopha.2019.109231). Thus, we used MLA to block α7nAChR in this colitis model.

3.Does blocking of α7nAChR affects autophagy in mouse colon?

Re: We found that knockout α7nAChR in mouse colon resulted in decreased autophagy, as indicated by decreased LC3Ⅱ/LC3Ⅰ ratio (Figure S5E-F).

4.It will be beneficial to add some information on α7nAChR in the discussion on what type of cells express α7nAChR and its physiological relevance and function, and possible mechanisms underlying its regulation by autophagy.

Re: Thank you for your professional suggestion. We have added the concerns in the Discussion in the revised manuscript (Line 355-371).

---

## [Editor Report · Decision Letter 2]

1 Sep 2023

Combined Administration of Anisodamine and Neostigmine Alleviated Colitis by Inducing Autophagy and Inhibiting Inflammation

PONE-D-22-21418R2

Dear Dr. Ke,

We’re pleased to inform you that your manuscript has been judged scientifically suitable for publication and will be formally accepted for publication once it meets all outstanding technical requirements.

Kind regards,

Pradeep Dudeja

Academic Editor

PLOS ONE
---

## [Editor Report · Acceptance letter]

5 Feb 2024

PONE-D-22-21418R2 

PLOS ONE

Dear Dr. Ke, 

I'm pleased to inform you that your manuscript has been deemed suitable for publication in PLOS ONE. Congratulations! Your manuscript is now being handed over to our production team.

Kind regards, 

on behalf of

Dr. Pradeep Dudeja 

Academic Editor

PLOS ONE